# Proteomic Profile of *Daphnia pulex* in Response to Heavy Metal Pollution in Lakes of Northern Patagonia

**DOI:** 10.3390/ijms26010417

**Published:** 2025-01-06

**Authors:** Juan-Alejandro Norambuena, Patricia Poblete-Grant, Jorge F. Beltrán, Patricio De los Ríos-Escalante, Cristian Aranzaez-Ríos, Jorge G. Farías

**Affiliations:** 1Ph.D. Program on Natural Resources Sciences, Universidad de La Frontera, Avenida Francisco Salazar, 01145, P.O. Box 54-D, Temuco 4811230, Chile; 2Department of Chemical Engineering, Faculty of Engineering and Science, Universidad de La Frontera, Avenida Francisco Salazar, 01145, P.O. Box 54-D, Temuco 4811230, Chile; beltran.lissabet.jf@gmail.com (J.F.B.); c.aranzaez01@ufromail.cl (C.A.-R.); 3Department of Biological and Chemical Sciences, Faculty of Natural Resources, Catholic University of Temuco, Manuel Montt, 56, P.O. Box 15-D, Temuco 4813302, Chile; prios@uct.cl; 4Centre of Plants, Soil Interaction and Natural Resources Biotechnology, Scientific and Biotechnological Nucleus (BIOREN), Universidad de La Frontera, Avenida Francisco Salazar, 01145, P.O. Box 54-D, Temuco 4811230, Chile; patricia.poblete@ufrontera.cl; 5Nucleus of Environmental Studies, UC Temuco, Manuel Montt, 56, P.O. Box 15-D, Temuco 4813302, Chile

**Keywords:** *Daphnia pulex*, Northern Patagonian lakes, anthropogenic impacts, heavy metals, proteomic analysis

## Abstract

Over recent decades, Northern Patagonia in Chile has seen significant growth in agriculture, livestock, forestry, and aquaculture, disrupting lake ecosystems and threatening native species. These environmental changes offer a chance to explore how anthropization impacts zooplankton communities from a molecular–ecological perspective. This study assessed the anthropogenic impact on *Daphnia pulex* by comparing its proteomes from two lakes: Llanquihue (anthropized) and Icalma (oligotrophic). Results showed substantial differences in protein expression, with 17 proteins upregulated and 181 downregulated in Llanquihue, linked to elevated levels of copper, manganese, dissolved solids, phosphate, and nitrogen. These stressors caused metabolic damage and environmental stress in *D. pulex*. Our findings highlight the importance of monitoring pollution’s effects on Northern Patagonian ecosystems, especially on keystone species like *D. pulex*, essential for ecosystem stability. This research provides fresh molecular–ecological insights into pollution’s impacts, a perspective rarely addressed in this region. Understanding these effects is critical for conserving natural resources and offers pathways to study adaptive mechanisms in keystone species facing pollution. This approach also informs strategies for ecosystem management and restoration, addressing both immediate and long-term challenges in Northern Patagonian aquatic environments.

## 1. Introduction

Freshwater ecosystems are essential for global biodiversity, providing habitats for a wide range of organisms and ecosystem services crucial for human well-being [1,2]. However, these ecosystems face increasing threats from pollution, particularly heavy metals, which represent a significant global environmental issue [3]. Anthropogenic activities such as mining, industry, agriculture, and urbanization have contributed to the accumulation of heavy metals, including lead, cadmium, mercury, copper, and manganese, in freshwater bodies [4,5,6]. These pollutants cause long-lasting toxic effects, such as bioaccumulation within trophic networks and the disruption of critical ecological functions, including those performed by zooplankton communities [7,8,9,10].

Zooplankton communities play a fundamental role in freshwater aquatic ecosystems, transferring energy captured by primary producers like microalgae to higher trophic level species such as fish [11]. Among the diverse zooplankton in Chilean Northern Patagonian lakes, there is a Cladocera species, *Daphnia pulex* (*D. pulex*) Leydig, 1860 [4]. *D. pulex* primarily feeds on microalgae, abundant in mesotrophic or eutrophic lakes. In Northern Patagonia, lakes, such as Lake Llanquihue, with high nutrient and contaminant levels, contrast sharply with oligotrophic lakes like Lake Icalma, characterized by low nutrient concentrations and relatively pristine conditions [4,9,12].

Lakes in Chilean Northern Patagonia, characterized by low nutrient levels (oligotrophic), were classified based on their morphology, physicochemical properties, biodiversity, and environmental conditions during the 1980s and 1990s [4,13,14,15,16]. However, in recent decades, the expansion of agriculture, livestock, forestry, and aquaculture has led to increased soil and water pollution, largely due to intensive and unsustainable resource management practices [17,18,19,20]. Lake Llanquihue is notably affected by heavy metal pollution, including copper and manganese, primarily from agricultural runoff, aquaculture activities, and urban discharge [4,21]. In contrast, Lake Icalma remains relatively untouched, serving as an ideal reference site for studying natural ecosystems [4].

Currently, several tools are available for assessing anthropogenic impacts on freshwater ecosystems [4,22,23]. One effective approach is using bioindicator species, such as *D. pulex*, extensively studied for its sensitivity to environmental stressors through ecological and ecotoxicological methods [24,25]. In addition, *D. pulex* has become a model species in molecular research, including environmental genomics, proteomics, and epigenetics. Its phenotypic plasticity in response to environmental changes has driven significant research into this organism, highlighting its value in scientific studies [26,27,28,29].

Environmental proteomics has proven crucial in linking protein diversity to ecological functions in aquatic ecosystems [30]. While an organism’s genome establishes its inherent traits, proteins drive the dynamic and adaptive processes essential for survival in changing environments. In aquatic ecosystems, proteins undergo specific modifications in response to environmental shifts, making proteomics an indispensable tool for exploring these systems [30,31].

Research on protein expression in aquatic ecosystems impacted by human activities has been relatively limited, despite its relevance for understanding how these activities affect species. In this study, a proteomic approach was used to evaluate the responses of *D. pulex* to different levels of heavy metal pollution in two Northern Patagonian lakes with contrasting water qualities. Comparing these unique and fragile ecosystems provides valuable information on the effects of human activities on biodiversity and ecological functions. This work highlights the need to integrate molecular and ecological approaches to advance understanding and mitigation of aquatic biodiversity loss, providing an essential perspective for the conservation of these freshwater ecosystems.

## 2. Results

### 2.1. Protein Extraction and Proteomics

The protein concentrations in *D. pulex* from the two Northern Patagonian lakes under study were measured as 150.41 µg/mL^−1^ for Icalma and 152.23 µg/mL^−1^ for Llanquihue. Before the proteomic analysis, the samples were adjusted to a uniform concentration of 150 µg/mL^−1^. Proteomic analysis identified a total of 1247 proteins (Figure 1), with 17 significantly (*p* ≤ 0.05) upregulated proteins (Table 1) and 181 downregulated proteins (Table 2) in Llanquihue compared to Icalma. Of these 181 downregulated proteins, only 6 were analyzed in this study, selected for their roles in oxidative stress, reactive oxygen species (ROS), decreased ATP production, and exoskeleton stability.

The upregulated proteins in individuals of *D. pulex* collected in Llanquihue were involved in the response of this species to environmental stress, including calcium-transporting ATPase, EV-type proton ATPase subunit E, tubulin alpha chain, two variants of heat shock 70 kDa protein cognate 4, fructose-bisphosphate aldolase, heat shock protein 83, 90, and superoxide dismutase. These proteins were more abundant in individuals from Llanquihue, being associated with the muscular system (e.g., myosin regulatory light chain), carbohydrate metabolism (e.g., glyceraldehyde-3-phosphate dehydrogenase, isocitrate dehydrogenase [NADP]), and physiological processes such as ovarian maturation (vitellogenin) (Table 1). In contrast, the 181 downregulated proteins included those associated with the response to environmental stressors (e.g., cytochrome C oxidase subunit 5A, cytochrome C oxidase subunit 2, NADH ubiquinone oxidoreductase 75 kDa subunit, NADH dehydrogenase [ubiquinone] flavoprotein 1), chitin-related proteins (chitin-binding type-2), and proteins involved in energy metabolism (ATP synthase subunit gamma) (Table 2).

### 2.2. Relationships of Up- and Downregulated Proteins with Physicochemical and Ecological Factors in Northern Patagonian Lakes

The proteomic profile of *D. pulex* was investigated in two contrasting environments: the oligotrophic lake Icalma and the anthropized lake Llanquihue. We observed that protein abundance was significantly influenced by total dissolved solids (TDS), calcium (Ca), total nitrogen (N), electrical conductivity (EC), manganese (Mn), pH, copper (Cu), and phosphate concentration (Figure 2). Downregulated proteins negatively correlated with TDS, EC, heavy metal concentration of Mn and Cu, total N, and phosphate. Conversely, upregulated proteins positively correlated with TDS, total N, EC, phosphate, and concentrations of Mn, Cu, and iron (Fe). However, these upregulated proteins were negatively correlated with pH, Ca concentration, and ecological parameters such as specific abundance (Ni’) and evenness (J’) diversity indicators.

PCA results indicated that PC1 accounted for 94.3% and PC2 for 2.6% of the data variability (Figure 3). The proteomic and chemical profiles distinctly separated the two lakes. Llanquihue showed a positive correlation with upregulated proteins and chemical parameters (phosphate, total N, Cu, Mn, TDS, EC) but a negative correlation with downregulated proteins, ecological parameters (Ni’ and J’), and other chemical variables (pH, temperature, Ca).

## 3. Discussion

Proteomics studies provide qualitative and quantitative information about the cell and tissue proteins of freshwater species under anthropogenic pressure by identifying molecular markers of the differential protein expression as a result of the effect of xenobiotic elements in the aquatic environment [36,52,53,54]. It is important to emphasize from the outset that, given the breadth of information generated in this study, we will only discuss certain proteins whose evidence has been demonstrated in the scientific literature due to their association with the response of different organisms to anthropized environments, which would illustrate the robustness of this type of analysis. However, all the data from this study will be freely available to anyone interested in evaluating potential markers of anthropization.

This study showed evidence of environmental stress due to economic and human activities in *D. pulex*. Our findings indicated an upregulation of the calcium-transporting protein ATPase, known in *D. pulex* and associated with heavy metals, including Cu, dissolved in a polluted environment [32]. The significant upregulation of this protein (Table 1) suggested a potential response to elevated Cu concentrations, similar to the report by Liorti et al. [32] regarding Lake Ontario. In addition, we found that the EV-type proton ATPase protein was upregulated in individuals of *D. pulex* collected in Llanquihue. This protein was involved in similar cellular processes [33], and it has been demonstrated to be involved in heavy metal tolerance in several species, like *Saccharomyces cerevisiae* [55], *Tamarix hispida* [56], and other plant species like *Mesembryanthemum crystallinum* [57]. Furthermore, it was demonstrated that *Cucumis sativus* plants treated with high concentrations of Cu and nickel (Ni) induced a pronounced upregulation of certain transcript isoforms encoded by this ATPase gene. Interestingly, the study concluded that the isoforms CsVHA-c1, CsVHA-c2, and CsVHP1;1 were essential elements in the mechanisms involved in the adaptation of cucumber plants to Cu toxicity [58]. Although there is a distinct evolutionary lineage between the two species, the overexpression of this protein in *D. pulex* (Table 1, Figure 1 and Figure 2) and *Cucumis sativus* suggests a common evolutionary defense mechanism for tolerating environments contaminated with heavy metals.

One of the most important reactions in glycolysis involves glyceraldehyde-3-phosphate dehydrogenase (GAPDH), where it breaks down glucose to obtain energy and carbon molecules [59]. Studies have observed that GAPDH plays a crucial role in the adaptation and tolerance of plants and aquatic organisms in contaminated environments, highlighting that exposure to high concentrations of heavy metals induces the production of reactive oxygen species (ROS). Among enzymatic responses, GAPDH is overexpressed as part of the antioxidant and detoxification response in plants and aquatic organisms. This overexpression is vital for managing oxidative stress and protecting cells from damage [60], which is consistent with our results of significant upregulation of GAPDH (Table 1) in *D. pulex* individuals collected at the sampling site of Llanquihue.

Isocitrate dehydrogenase is an important enzyme in the Krebs cycle, as it catalyzes the conversion of isocitrate to alpha-ketoglutarate, producing NADPH or NADH in the process [61]. A study that analyzed the stress state in *Rana sylvatica* found increased enzyme activity and NADPH production associated with the upregulation of isocitrate dehydrogenase, suggesting that this behavior may enhance antioxidant activity and defense against oxidative stress [62]. The analysis carried out in our work similarly showed increased activity of isocitrate dehydrogenase, which reflects the significant state of oxidative stress found in *D. pulex* individuals collected in Llanquihue.

One of the most important functions of the cell cytoskeleton involves the alpha-tubulin chain, which forms part of the microtubules essential for cell division, intracellular transport, and cell motility [63]. Studies have shown that the alpha-tubulin chain plays a crucial role in the adaptation and tolerance of *D. pulex* in aquatic environments contaminated by heavy metals and the quality of food resources [37,64]. Exposure to high concentrations of heavy metals induced the production of reactive oxygen species (ROS). Thus, the overexpressed alpha-tubulin chain found in this study may be explained as part of the antioxidant and detoxification response in *D. pulex* individuals. This overexpression is crucial for managing oxidative stress and protecting cells from ROS damage [64,65]. In addition, it has been observed that exposure to a common heavy metal can mediate several key life history responses in *D. pulex*, including somatic growth rate and survival rates [64]. These findings suggested a protective role of the alpha-tubulin chain in the adaptation and survival of *D. pulex* under the high concentrations of heavy metals found in Llanquihue.

Heat shock proteins are essential as molecular chaperones, enabling the correct folding of recently synthesized and misfolded proteins resulting from cellular stress factors. These proteins were chosen as bioindicators for the early detection of cellular distress due to their significance in cellular functions [66]. Expression levels of the 70 kDa heat shock protein (HSP70) have been reported to increase linearly in the presence of heavy metals like cadmium, arsenic, nickel, and copper [67,68]. The results of HSP70 upregulation in this study (Table 1, Figure 1 and Figure 2) agree with the previously mentioned cellular response of organisms subjected to heavy metal concentrations. Consequently, the heavy metal concentrations in Llanquihue might have triggered the upregulation of this protein as a defense mechanism in *D. pulex* individuals.

In addition, we found a significant upregulation of superoxide dismutase (Table 1) in Llanquihue, which can be attributed to the high Cu concentrations previously reported in this lake [4], and which could constitute a powerful biomarker in these contaminated conditions. Superoxide dismutase is a potent antioxidant important in cellular defense against oxidative stress. This protein has several properties, such as a high rate of catalysis of reactions, as well as a high rate of stability against physicochemical stress [69]. Our results were in accordance with those found by Lyu et al. [70], where *D. magna* was exposed to high concentrations of Cu/zinc, obtaining that the expression of superoxide dismutase mRNA increased significantly (upregulated) after 48 h of exposition to high concentrations of Cu. They concluded that this gene constitutes a biomarker of oxidative stress for this heavy metal, and it was demonstrated that this enzyme exhibited a high sequence similarity of 88% with the *D. pulex* species.

At the same time, our study also indicated some downregulation in certain protein expressions (Table 2). *D. pulex* individuals collected from Llanquihue exhibited a downregulation of the cytochrome C oxidase protein subunits (cytochrome C oxidase subunit 5A, cytochrome C oxidase subunit 2). The above was observed in the anthropized lake Llanquihue, which has been reported to have high concentrations of Cu and Mn [4]. These metals can compete with oxygen at COX active sites, thus inhibiting the functioning of this complex and preventing the generation of the proton gradient. Consequently, ATP synthase cannot function efficiently, decreasing ATP synthesis and energy generation in *D. pulex* cells [51]. In organisms exposed to high concentrations of metals, downregulation of cytochrome oxidase may function as an adaptive defense mechanism [71]. This decrease in enzyme activity may enable cells to conserve energy and reduce the production of reactive oxygen species (ROS), thus mitigating oxidative damage caused by exposure to environmental pollutants such as copper and cadmium metals, which may be beneficial for cell survival under prolonged stress, as observed by Niemuth et al. [50]. This is supported by the fact that cytochrome oxidase subunits (COX) are known as part of complex IV, being a key component of the electron transport chain located within the inner plasma membrane of the mitochondria [71,72,73]. Thus, it plays a pivotal role in translocating protons across the membrane, thereby creating an electrochemical gradient that the ATP synthase enzyme utilizes to synthesize adenosine triphosphate (ATP), a fundamental source of energy [71,72,73]. In addition, cytochrome C oxidase activity could be observed to be downregulated by various mechanisms. For example, oxygen availability and the presence of specific inhibitors, such as high concentrations of metals, can affect its function, as explained by Muyssen et al. [48] and Ukhueduan et al. [49].

Likewise, we found that *D. pulex* individuals from Llanquihue exhibited a downregulation of the NADH ubiquinone subunits (NADH ubiquinone oxidoreductase 75 kDa subunit, NADH dehydrogenase [ubiquinone] flavoprotein 1). In the context of the electron transport chain, these complex I subunits facilitate the transfer of electrons from NADH to the Q coenzyme, resulting in the translocation of protons across the inner mitochondrial membrane. This process contributes to the creation of an electrochemical gradient that, in turn, enables the enzyme ATP synthase to synthesize ATP and produce energy [72]. Our findings align with the research conducted by Niemuth et al. [50] and Ukhueduan et al. [49], who conducted ecotoxicological experiments under controlled conditions to study the effects of heavy metal toxicity on *Daphnia*. Accordingly, they highlighted that cellular metabolism involving NADH served as a natural source of balanced ROS production. However, a high concentration of Cu and Mn in aquatic environments could lead to an overproduction of ROS, thereby disrupting the delicate balance within the mitochondria.

In the case of *D. pulex* individuals developed in lakes raised under anthropogenic conditions (Llanquihue), a downregulation of the ATP synthase subunit gamma protein was observed. This downregulation suggested a potential depletion of mitochondrial ATP, which could result in organelle dysfunction due to increased endogenous ROS production, consequently affecting various physiological processes [51,74]. Future studies could benefit from assessing the mitochondrial structure of *D. pulex* under environmental stress to further support the mitochondrial dysfunction hypothesis.

In Llanquihue, *D. pulex* individuals exhibited carapace instability coinciding with the downregulation of specific proteins such as chitin-binding type-2. This phenomenon aligns with prior findings by Otte et al. [39] and Becker et al. [37], who identified inadequate food quality or exposure to environmental stress as potential contributing factors to stress-induced situations that impacted chitin production in *Daphnia*. Chitin is an important polysaccharide of the cuticle, exoskeleton, and other structures in arthropods like *D. pulex* [4]. This molecule is essential for forming cell walls in plant cells and in the exoskeleton of arthropods such as *D. pulex*, providing them with physical protection in the environment where they develop [72]. Furthermore, our results from previous studies revealed significantly lower Ca concentrations in Llanquihue than in Icalma [4]. Given the essential role of Ca in the formation of invertebrate exoskeletons, these reduced levels may negatively impact calcium-demanding zooplankton crustaceans like *D. pulex* [75]. *D. pulex*, due to its periodic molting, exhibits a high demand for Ca, and reductions in Ca concentrations can lead to decreased reproduction and body size. During the sampling in this study, daphnids collected from Llanquihue appeared smaller in body size than those from Icalma. Although quantitative differences were not measured in our study, we consider that they must be included in future studies after the results obtained. Consequently, *D. pulex* may need to increase energy consumption to enhance Ca absorption, reallocating energy for growth. This phenomenon could be closely associated with the downregulated protein abundance of chitin-binding type-2, a protein involved in the morphological changes of the carapace [39]. Importantly, this protein’s abundance positively correlated with Ca concentration (Figure 2), further supporting this mechanism.

Furthermore, the notably elevated concentration of total dissolved solids (TDS) previously reported in Llanquihue [4] had a significant influence on the up- and downregulation of proteins in *D. pulex* (Figure 2). Accordingly, Chapman et al. [76] stated that the toxicity of TDS in freshwater ecosystems was primarily attributed to specific combinations and concentrations of ions (e.g., sodium, potassium, calcium, magnesium, chloride, sulfate, and bicarbonate). Moreover, Weber and Pirow [77] reported that *D. pulex* was physiologically sensitive to changes in water ion balance, affecting its ion and osmoregulatory processes, which is in accordance with the observed protein expression associated with high TDS concentration in our study (Figure 2).

Finally, our hypothesis can be supported due to the high data variability explained by PCA (96.9%) (Figure 3), signifying that freshwater quality affected the development of *D. pulex* at the molecular level, particularly in the anthropized lake Llanquihue. This impact can be largely attributed to TDS, EC, nutrient levels (P and N), and heavy metal concentrations. Ecological parameters were also considered due to their substantial contribution to data variability.

## 4. Materials and Methods

### 4.1. Study Area and Sampling

The populations of the water flea, *D. pulex* in Lake Icalma (La Araucanía region) and Lake Llanquihue (Los Lagos region), two Northern Patagonian lakes, were compared (Figure 4). Lake Icalma (38°48′ S and 71°17′ W) is classified as an oligotrophic lake with no anthropogenic interference. Lake Llanquihue (41°08′ S and 72°47′ W), under intense anthropogenic pressure, is classified as mesotrophic due to high nutrient inputs from agriculture, livestock, forestry, and aquaculture [4,9,13,17,78,79,80].

Molecular profiling of *D. pulex* individuals from the two Northern Patagonian lakes was conducted in March 2022. This month has been reported to be one of the periods of highest abundance of *D. pulex* [81]. Zooplankton samples, including *D. pulex*, were taken at sampling points I3 at Icalma (38°48′21″ S; 71°17′0.7″ W) and point LL3 at Llanquihue (41°19′17.5″ S; 72°57′53.5″ W) (Figure 5). The collection procedure was repeated until 60 *D. pulex* individuals were obtained at each site. Zooplankton samples were collected at a depth of 20 m using a Nansen net (Hydro-Bios; Altenholz, Schleswig-Holstein, Germany) 20 cm in diameter and a 200 µm mesh opening following the methodology described by De los Ríos-Escalante [81] and Woelfl et al. [82].

### 4.2. Physicochemical Properties of Study Sites

The physicochemical data utilized in this study were obtained from measurements conducted in parallel during a study previously published by Norambuena et al. [4]. In that study, water samples were collected from the same sampling points (I3 in Lake Icalma and LL3 in Lake Llanquihue) to evaluate parameters such as total dissolved solids (TDS), calcium (Ca^2+^), iron (Fe), manganese (Mn), copper (Cu), total nitrogen and phosphorus, temperature, and electrical conductivity (EC). These variables were selected due to their relevance in characterizing lacustrine environments and their potential influence on biological communities.

The analyses were conducted following standardized protocols described in detail in Norambuena et al. [4]. Water samples were collected using a Van Dorn device at 10 m depth and stored at 4 °C until laboratory analysis. Dissolved metal concentrations were identified using a Hanna Hi801 UV-Vis spectrophotometer at 340 to 900 nm range according to the methodologies detailed, while general parameters such as EC, TDS, and temperature were measured using WTW Multi 340i multiparameter probes, following the Standard Methods for the Examination of Water and Wastewater (APHA, AWWA, and WEF) [83]. These data were incorporated into the present study to establish relationships between environmental conditions and the proteomic profiles of *D. pulex*.

### 4.3. Protein Extraction and Quantification

*D. pulex* samples were collected from lakes from December 2021 to March 2022 in Icalma and Llanquihue, and transported to a cooler at 4 °C. Oxygen was provided to keep the individuals alive. Once in the laboratory, protein extraction was performed instantaneously. To obtain the proteome of the species, cell membrane protein extraction procedures were carried out using the commercial Mem-PER™ Plus Membrane Protein Extraction Kit (Thermo Fisher Scientific™; Waltham, MA, USA), selecting protocol number two, recommended for cell suspensions. *D. pulex* specimens were captured from integrated samples at site I3 in Icalma and site LL3 in Llanquihue for molecular protein expression and identification analysis. The protocol involved re-suspending cells in 1.5 mL of cell washing solution (CLS), transferring them to 2 mL vials, centrifuging at 300× *g* for 5 min, discarding the supernatant, adding 0.75 mL of permeabilization buffer (TP) to the pellet, and homogenizing by vortexing, followed by incubating the suspensions for 20 min at 4 °C with constant agitation. After constant shaking of the suspension of permeabilized cells, centrifugation was performed for 15 min at 16,000× *g*, and the supernatant (cytosolic proteins) was carefully removed and discarded. Subsequently, 0.5 mL of solubilization buffer (TS) was added to the obtained pellet, homogenized by pipetting and incubated at 4 °C for 30 min with constant agitation. After this, the samples were centrifuged at 16,000× *g* for 15 min at 4 °C, then the supernatant (membrane proteins) was carefully collected and stored at −80 °C until use. Protein concentration was determined by the bicinchoninic acid (BCA) assay using the Pierce™ BCA Protein Assay Kit (Thermo Fisher Scientific™; Waltham, MA, USA).

### 4.4. Proteomic Analysis

#### 4.4.1. Chemicals and Instrumentation

Iodoacetamide (IAA), DL-dithiothreitol (DTT), acetonitrile (ACN), and formic acid (FA) were purchased from Sigma (St. Louis, MO, USA), while trypsin (bovine pancreas) was purchased from Promega (Madison, WI, USA). Ultrapure water was prepared using a Millipore purification system (Billerica, MA, USA). An Ultimate 3000 nano UHPLC system was coupled to an ESI nanospray source to a Q Exactive HF mass spectrometer (Thermo Fisher Scientific™; Waltham, MA, USA).

#### 4.4.2. Sample Information

Total protein extracts from groups I3 (60 individuals) and LL3 (60 individuals) were digested with trypsin, identified, and quantified using a nanoLC-MS/MS platform.

#### 4.4.3. Sample Preparation

The sample buffer was exchanged with ammonium bicarbonate, and the samples had a final concentration of 1 μg/μL. Then, 60 μL of the sample was transferred to a new Eppendorf tube. After reduction with DTT (10 mM, 56 °C, 1 h) and alkylation with IAA (20 mM, room temperature in the dark, 1 h), the samples were centrifuged (12,000 rpm, 4 °C, 10 min) and washed once with 50 mM ammonium bicarbonate. Free trypsin was added to the protein solution in a trypsin-to-protein ratio of 1:50, along with 50 mM ammonium bicarbonate (100 μL), and the mixture was incubated overnight at 37 °C. Finally, the samples were centrifuged at 12,000 rpm at 4 °C for 10 min. Then, 100 μL of 50 mM ammonium bicarbonate was added to the device and centrifuged, and this step was repeated once. The extracted peptides were lyophilized to dryness and resuspended in 20 μL of 0.1% formic acid in preparation for LC-MS/MS analysis.

#### 4.4.4. NanoLC

Nanoflow UPLC: Ultimate 3000 nano UHPLC system (ThermoFisher Scientific, USA); nanocolumn: trapping column (PepMap C18, 100 Å, 100 μm × 2 cm, 5 μm) and an analytical column (PepMap C18, 100 Å, 75 μm × 50 cm, 2 μm); loaded sample volume: 1 μg; mobile phase: A: 0.1% formic acid in water; B: 0.1% formic acid in 80% acetonitrile. Total flow rate: 250 nL/min; LC linear gradient: from 2% to 8% buffer B in 3 min, from 8% to 20% buffer B in 56 min, from 20% to 40% buffer B in 37 min, and finally from 40% to 90% buffer B in 4 min.

#### 4.4.5. Mass Spectrometry

The full scan was conducted from 300 to 1650 *m*/*z* at a resolution of 60,000 at 200 *m*/*z*, with an automatic gain control target of 3 × 10^6^. The MS/MS scan was performed in Top 20 mode, which involved selecting the top 20 precursor ions for fragmentation, using the following parameters: a resolution of 15,000 at 200 *m*/*z*, an automatic gain control target of 1 × 10^5^, a maximum injection time of 19 ms, normalized collision energy at 28%, an isolation window of 1.4 Th, and dynamic exclusion for 30 s.

#### 4.4.6. Proteome Data Analysis

The six raw MS files were analyzed and searched against the *D. pulex* protein UniProt database, corresponding to the species of the samples, using MaxQuant (version 1.6.2.6). The protein modification parameters included cysteine (C) carbamidomethylation as a fixed modification and methionine (M) oxidation as a modification variable. Enzyme specificity was set to trypsin, allowing for up to two missed cleavages. The precursor ion mass tolerance was set at 10 ppm and the MS/MS tolerance at 0.6 Da.

### 4.5. Statistical Analysis

To compare the expression of proteins in the two lakes, Icalma and Llanquihue, a paired *t*-test with a significance level of *p* ≤ 0.05 was used in all cases. Statistical analyses for comparison were performed using the Microsoft Excel package (Microsoft Office 365). Protein profile visualization was carried out using the Python programming language and the Matplotlib library. Protein profile visualization was conducted using the Python programming language (https://www.python.org/; accessed on 3 April 2024) in conjunction with the Matplotlib library (https://matplotlib.org/; accessed on 3 April 2024).

The data obtained, including physicochemical measurements from a previous study [4], and protein intensity (label-free quantification, LFQ) were subjected to a check for normality using the Shapiro–Wilk test and for homogeneity of variance using the Levene test. Significant differences were analyzed using a parametric two-way ANOVA at a 95% significance level, followed by post-hoc Tukey’s honestly significant difference (HSD) test. Detected significant relationships were further analyzed using Pearson’s correlation analysis, with significance accepted at *p* ≤ 0.05. A principal component analysis (PCA) was performed using the Factoextra package in the R software (version 4.3.2) to identify the variables that explained the variability in the data. All statistical tests were conducted using the R Foundation for Statistical Computing, Version 3.6.3 (R Development Core Team, 2009–2018).

## 5. Conclusions

Our study showed that anthropogenic pressure significantly impacted the proteome of *Daphnia pulex*, resulting in the differential expression of 17 significantly upregulated and 181 downregulated proteins. Of these 198 proteins, only 13 were analyzed, evidencing a significant anthropogenic pressure in Llanquihue affecting zooplankton like *D. pulex* individuals compared to those collected in Icalma. The observed up- and downregulated proteins indicate cellular stress, compromising physiological functions, particularly cellular metabolism and exoskeleton composition. This study provides valuable information on the effects of anthropogenic pressure on freshwater ecosystems and highlights the importance of incorporating a molecular–ecological approach into environmental monitoring and management strategies. Further studies on the genetic divergence of the precursor genes of the tested proteins will shed light on the influence of environmental change due to anthropogenic stress on gene expression. Our findings have implications for the conservation and restoration of freshwater ecosystems by identifying which anthropogenic variables are affecting zooplankton species and how this stress affects them. This will effectively facilitate the advancement of sustainable, eco-friendly management practices to support human economic activity in this region and prevent premature pollution in the lakes of Chilean Northern Patagonia.

## Figures and Tables

**Figure 1 ijms-26-00417-f001:**
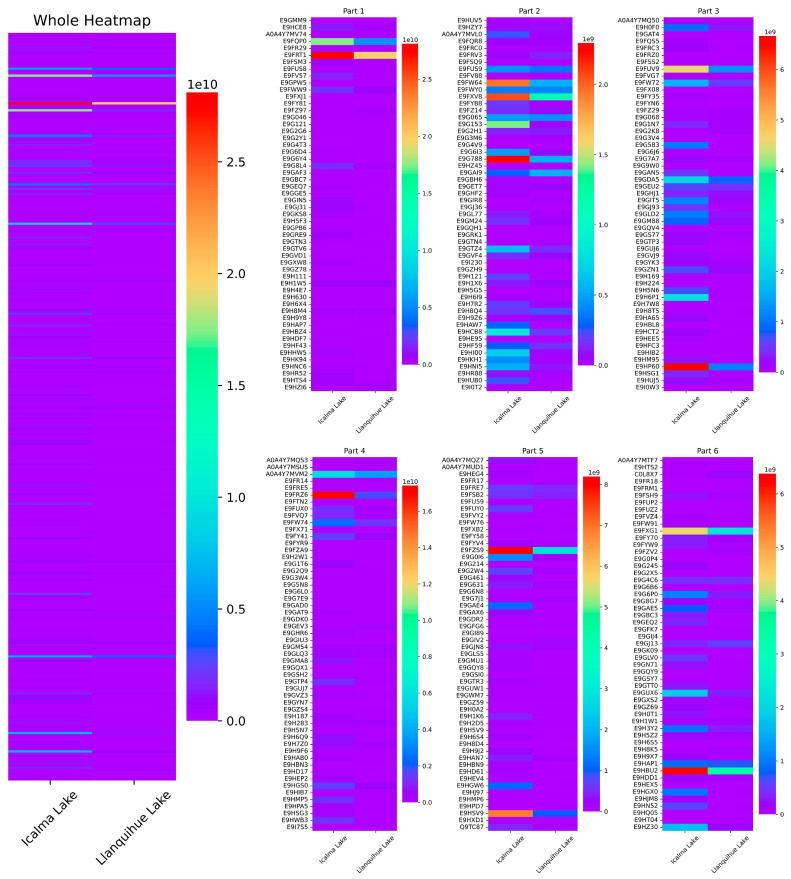
Differential protein expression in *D. pulex* from lakes at Icalma (left column) and Llanquihue (right column). The total proteins are displayed in a comprehensive heat map, subdivided into six parts (Part 1, 2, 3, 4, 5, and 6) for a detailed comparison of their differences.

**Figure 2 ijms-26-00417-f002:**
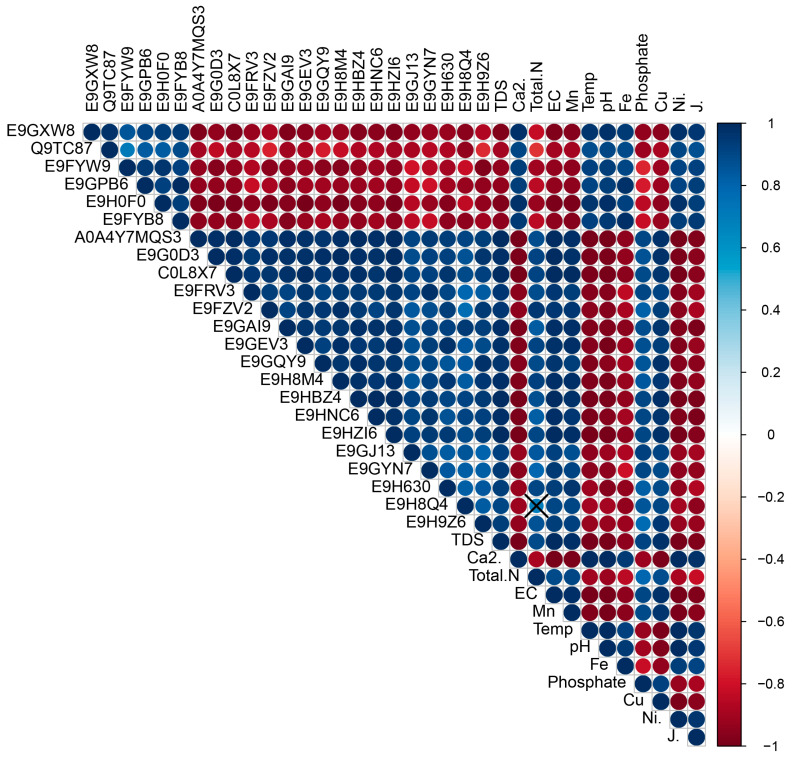
Correlation matrix of protein expression values in relation to various physicochemical (total dissolved solids [TDS], dissolved calcium [Ca^2+^], total nitrogen, electrical conductivity [EC], manganese [Mn], temperature, dissolved iron [Fe], dissolved copper [Cu]) and ecological variables (specific abundance [Ni], evenness [J’]) at sampling sites in Lakes Icalma and Llanquihue. Proteins are labeled using FASTA numbers. Positive correlations are denoted by blue circles, while negative correlations are indicated by red circles. Circles without a cross represent significant correlations (*p* ≤ 0.05) as determined by Pearson’s test.

**Figure 3 ijms-26-00417-f003:**
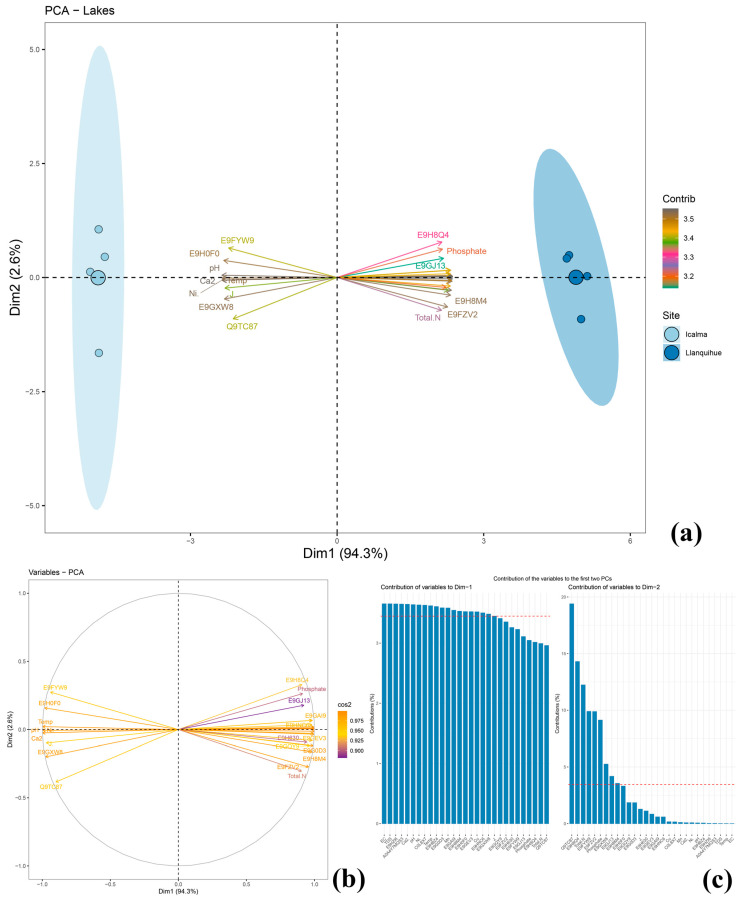
PCA of protein expression, labeled using FASTA numbers, alongside various physicochemical (total dissolved solids [TDS], dissolved calcium [Ca^2+^], total nitrogen, electrical conductivity [EC], manganese [Mn], temperature, dissolved iron [Fe], dissolved copper [Cu]) and ecological variables (specific abundance [Ni], evenness [J’]), measured in Lakes Icalma and Llanquihue. The figure includes a PCA biplot (**a**), a biplot showing only the variables (**b**), and the contribution of these variables to the PCA (**c**). It explains 94.3% of the variance in PC1 and 2.6% in PC2 axes.

**Figure 4 ijms-26-00417-f004:**
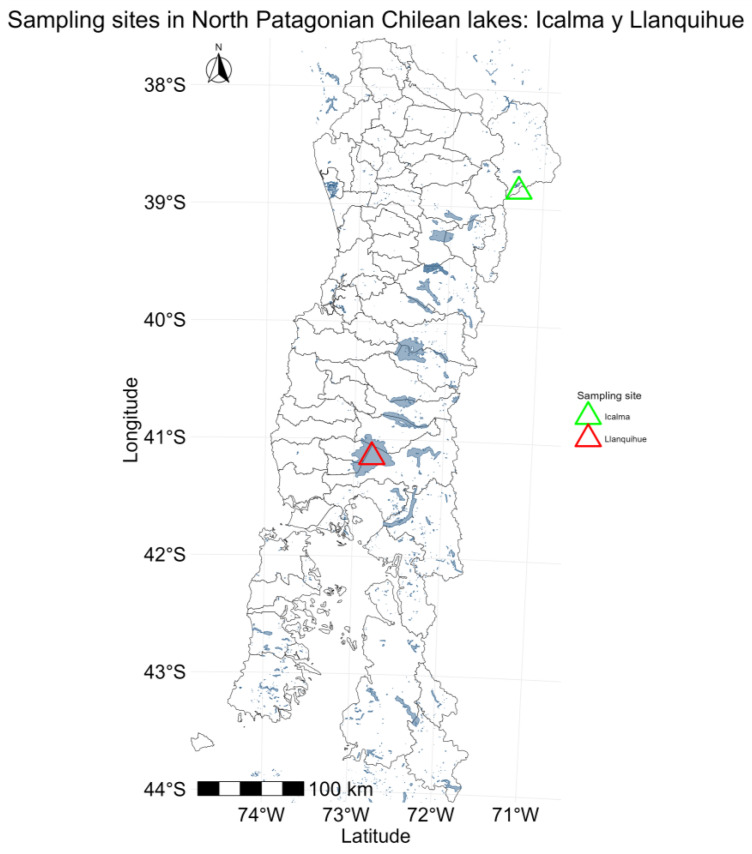
Map of the Northern Patagonian area of Chile, 38° S, 71° W and 41° S, 72° W, encompassing the regions of La Araucanía and Los Lagos, Lake Icalma (green triangle), and Lake Llanquihue (red triangle). This area is notable for the presence of *D. pulex*.

**Figure 5 ijms-26-00417-f005:**
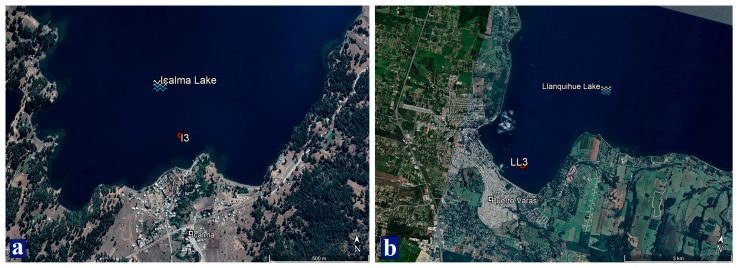
Sampling sites in Icalma (**a**) and Llanquihue (**b**) lakes. Image sourced from Google Earth Pro 7.3.2.5776.

**Table 1 ijms-26-00417-t001:** Upregulated proteins in *D. pulex* found in the anthropized lake of Llanquihue in the Northern Patagonian area. This table includes mean values and standard deviations (SD) of label-free quantitation (LFQ) for protein expression. It also shows significant difference values from a paired *t*-test (*p* < 0.05) and references related to the effects at the cellular level in *D. pulex*.

FASTA Code and Protein Name–*Daphnia pulex*	Mean LFQ Intensity Icalma Lake	SD	Mean LFQ Intensity Llanquihue Lake	SD	*t* Student	Effects of High Expression in *Daphnia pulex* Cells	Reference
tr|A0A4Y7MQS3|Calcium-transporting ATPase.	2,3049,333.33	19,962,075.30	296,400,000.00	7,180,619.75	0.0007	Stress to protection against high dissolved copper concentration.	[32]
tr|E9G0D3|EV-type proton ATPase subunit E.	102,897,333.33	10,175,529.93	181,240,000.00	2,155,643.755	0.00202	Stress due to pH differential in intra- and extra-cellular compartments.	[33]
tr|C0L8X7|Glyceraldehyde-3-phosphate dehydrogenase.	56,312,666.67	2,641,977.353	188,210,000.00	12,966,402.74	0.00195	Its expression depends on extracellular iron concentrations.	[34,35]
tr|E9FRV3|Isocitrate dehydrogenase [NADP].	70,594,333.33	13,921,006.23	129,763,333.33	5,658,801.40	0.01038	Reducing glutathione disulfide (GSSG) to GSH for antioxidant purposes.	[36]
tr|E9FZV2|Aldedh domain-containing protein.	45,473,666.67	7,792,250.92	77,216,333.33	6,030,186.924	0.00074	No information.	-
tr|E9GAI9|Tubulin alpha chain.	291,610,000.00	27,511,075.95	544,726,666.67	34,310,777.22	0.00728	Stress to changes in temperature and food supply.	[37]
tr|E9GEV3|Heat shock 70 kDa protein cognate 4.	140,560,000.00	6,211,094.91	289,060,000.00	29,163,051.97	0.00526	Stress to changes in temperature.	[38]
tr|E9GJ13|Fructose-bisphosphate aldolase.	373,083,333.33	31,550,152.67	572,193,333.33	67,951,014.95	0.02975	Stress related to metabolic processes.	[39]
tr|E9GQY9|40S ribosomal protein SA.	13,416,333.33	3,566,336.82	29,646,333.33	2,150,024.73	0.01879	Effects on Ribosomal RNA production.	[40,41]
tr|E9GYN7|Arginase.	30,657,666.67	3,614,230.95	45,341,333.33	2,336,978.03	0.00278	Effects on nitrogen metabolism, urea cycle.	[42]
tr|E9H630|Myosin regulatory light chain.	57,037,333.33	2,622,803.14	78,688,666.67	7,572,951.49	0.03065	Effect on muscle fibers.	[43,44]
tr|E9H8M4|14_3_3 domain-containing protein.	348,406,666.67	12,185,697.90	441,686,666.70	13,412,659.44	0.00161	Effects on diverse signaling proteins, including kinases, phosphatases, and transmembrane receptors.	[45]
tr|E9H8Q4|Vitellogenin.	169,703,333.33	24,323,168.65	230,780,000.00	13,865,067.62	0.0142	Effect on egg yolk precursor.	[37]
tr|E9H9Z6|40S ribosomal protein S13.	41,323,000	10,951,575.64	73,726,333.33	3,100,061.343	0.02748	Effects on Ribosomal RNA production.	[40,41]
tr|E9HBZ4|Heat shock 70 kDa protein cognate 4.	36,423,333.33	5,107,539.949	181,830,000.00	13,299,966.17	0.00265	Stress to changes in temperature.	[38]
tr|E9HNC6|Heat shock protein 83, 90.	177,840,000.00	6,062,309.79	279,873,333.33	18,034,783.98	0.00591	Intracellular stress in a climate change of the aquatic environment.	[46]
tr|E9HZI6|Superoxide dismutase.	226,633,333.33	4,314,583.80	514,356,666.67	8,345,719.46	0.00012	Stress response to copper, ammonia, and hypoxia levels.	[47]

**Table 2 ijms-26-00417-t002:** Downregulated proteins in *D. pulex* found in the anthropized lake of Llanquihue in the Northern Patagonian area. This table includes mean values and standard deviations (SD) of label-free quantitation (LFQ) for protein expression. It also presents significant difference values obtained from Student’s *t*-test (*p* < 0.05) and references related to the effects at the cellular level in *D. pulex*.

FASTA Code and Protein Name–*Daphnia pulex*	Subcellular Location	Mean LFQ Intensity Icalma Lake	SD	Mean LFQ Intensity Llanquihue Lake	SD	*t* Student	Effects of Down-Regulated in *Daphnia pulex*	Reference
tr|E9GXW8|Cytochrome C Oxidase subunit 5A.	Mitochondrion.	395,303,333.33	77,583,748.51	31,921,000.00	27,890,177.68	0.01	Oxidative Stress, ROS.	[48,49]
tr|Q9TC87|Cytochrome C Oxidase subunit 2.	Mitochondrion.	510,963,333.33	158,239,801.04	61,080,000.00	105,793,663.33	0.04	Oxidative Stress, ROS.	[48,49]
tr|E9FYW9|NADH Ubiquinone Oxidoreductase 75 kDa subunit.	Mitochondrion.	317,076,666.67	69,335,851.00	121,610,000.00	18,139,666.48	0.01	Multi-system disorders. Metal oxide nanomaterials, ROS.	[49,50]
tr|E9GPB6|NADH Dehydrogenase [Ubiquinone] flavoprotein 1.	Mitochondrion.	239,323,333.33	51,992,494.01	101,170,000.00	3,340,119.76	0.02	Multi-system disorders. Metal oxide nanomaterials, ROS.	[49,50]
tr|E9H0F0|ATP Synthase subunit gamma.	Mitochondrial proton-transporting ATP synthase complex.	939,796,666.67	66,471,600.95	293,256,666.67	113,510,292.63	0.01	Decreased ATP production.	[51]
tr|E9FYB8|Chitin-binding type-2.	Extracellular region.	179,673,333.33	23,052,571.08	34,680,000.00	60,067,522.01	0.047	Exoskeleton stability, stress to changes in temperature and food supply.	[37,39]

## Data Availability

For enquiries about information, please write to the corresponding authors.

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
