# Peer review of "Proteomic Profile of Daphnia pulex in Response to Heavy Metal Pollution in Lakes of Northern Patagonia"

_ijms, 2025, doi:10.3390/ijms26010417_

Round 1
Reviewer 1 Report
Comments and Suggestions for Authors
This study titled “Proteomic profile of Daphnia pulex in response to heavy metal pollution in lakes of Northern Patagonia” explored the impact of anthropogenic pressure on the freshwater cladoceran Daphnia pulex, a key species in aquatic ecosystems, by analyzing changes in its proteome in response to environmental stress. The research focuses on two lakes in Northern Patagonia, to investigate how human activities, such as urbanization and industrialization, affect Daphnia pulex. Using proteomics, the authors identify changes in protein expression linked to oxidative stress, metabolic pathways, and other biological processes, providing insights into the molecular mechanisms underlying stress responses. Major comments are as follows:
1- Introduction: please provide more details on how the two studied lakes are polluted and differ in their heavy metals pollution levels. Please describe why this study is important and highlight the novelty of your study.
2- Methods: Have you measured heavy metal levels? The study would benefit from including direct measurements of heavy metal concentrations in water or sediment samples from both lakes. This would strengthen the link between environmental stressors and the observed proteomic changes. Alternatively, the authors could discuss this limitation and suggest future studies to quantify heavy metal levels to confirm their role in driving proteomic variations.
3- Results: why we should consider the variation in proteomic is related only to heavy metal pollution? Without direct measurements of heavy metals, it is challenging to definitively attribute the observed proteomic variations in Daphnia pulex to heavy metal pollution. This limitation could weaken the study's conclusions, as other environmental factors (e.g., organic pollutants, eutrophication) might also contribute to the observed effects.
Author Response
Dear reviewer, we thank for your comments and your detailed review of the manuscript. Significant changes to the manuscript were made following your suggestions. The details of your suggestions are explained below:
1- Introduction: We are grateful for the reviewer's suggestion. The introductory section of the manuscript has been significantly improved. Details of the lakes studied have been provided in addition to a description of the importance and novelty of this study. Please see the first paragraph and the lines 49-51, 57-60, 69-73, and 74-82.
2- Methods: We appreciate the reviewer's comment and suggestion. The levels of heavy metals in water from lakes Icalma and Llanquihue were measured in a parallel study (Norambuena et al., 2022; Sustainability Journal; https://doi.org/10.3390/ su14106052). This study reported concentrations of copper, manganese and dissolved iron, among others, and concluded that Lake Llanquihue has significantly higher levels of these metals compared to Lake Icalma, reflecting a significant effect of anthropisation. We have included a summary of these physicochemical data in the main text to better contextualise our results.
3-Results: Thank you for pointing out this important observation. We recognise that proteomic variation could be influenced by multiple environmental factors, such as the presence of heavy metals and organic pollutants. We have incorporated additional information in the Methods and Discussion sections to clarify that the observed proteomic correlations are interpreted in the context of the physicochemical variables measured in a parallel study (Norambuena et al., 2022). This parallel study, published in Sustainability journal, measured dissolved heavy metal concentrations (Mn, Cu, Fe) along with other parameters, and these variables were used to establish the correlations reported in this manuscript.

Reviewer 2 Report
Comments and Suggestions for Authors
Manuscript ID ijms-3380322 entitled "Proteomic profile of Daphnia pulex in response to heavy metal pollution in lakes of Northern Patagonia" describes an examined the impact of anthropisation on Daphnia pulex by comparing its proteomes from two lakes: Llanquihue (anthropised) and Icalma (oligotrophic). Results showed significant differences in protein expression. The findings highlight the importance of monitoring pollution's effects on Northern Patagonian ecosystems, especially on keystone species like D. pulex, which are essential for ecosystem stability.
In the opinion of the reviewer, the article is interesting but needs some additions:
1. The introductory chapter should be extended. Namely, the authors should move from the general to the specific, i.e. a description of the problem of heavy metal pollution of lakes on a global scale is missing.
2. Although the authors present in Figure 2. Correlation matrix of protein expression values with different physicochemical (total dissolved solids [TDS], dissolved calcium [Ca2+], total nitrogen, electrical conductivity [EC], manganese [Mn], temperature, dissolved iron [Fe], dissolved copper [Cu]) and ecological variables (specific abundance [Ni], evenness [J']) at sampling sites in lakes Icalma and Llanquihue. These results can be included in the main text of the manuscript or as supplementary material.
3. Figure 1 "Figure 1. Differential protein expression in D. pulex from lakes at Icalma (left column) and 80 Llanquihue (right column)" needs improvement. The figure is currently illegible.
4. Please double-check the literature citations and the references.
Author Response
We are grateful for the reviewer's comments and suggestions. We are grateful for the reviewer's contribution and believe that all suggestions made greatly improved the quality of the manuscript. Please find below the response to each suggestion:
1. The introduction has been revised and modified according to the reviewer's suggestion. Please see first paragraph describing the problem of heavy metal pollution of lakes on a global scale. In addition, further modifications were made, according to other suggestions.
2. We are grateful for their suggestion on the integration of the correlation matrix results in the main text. We have included in the methodology an explicit reference to the physicochemical measurements obtained in the published parallel study (Norambuena et al., 2022; Sustainability Journal; https://doi.org/10.3390/su14106052), to provide a solid frame of reference for the proteomic correlations presented in Figure 2. These variables, measured through the parallel study, have been incorporated in the manuscript, and the main results are already discussed.
3. We acknowledge the reviewer’s comment regarding the legibility of Figure 1. The figure included in the main manuscript has been uploaded with the highest resolution possible in the manuscript submission system, which only allows for integration within the document. Unfortunately, the system does not support uploading figures independently. However, we have discussed this issue with the journal editor and have sent the original high-resolution *.tif file of Figure 1 directly to the editorial office. This file will be made available to the reviewers to ensure the figure’s quality and clarity for assessment.
4. We have reviewed bibliographic citations and references, which are shown in text and corresponding tables.

Round 2
Reviewer 1 Report
Comments and Suggestions for Authors
The authors responded to the suggestions and there are no other comments.